# Vector Competence of Mosquitoes from Germany for Sindbis Virus

**DOI:** 10.3390/v14122644

**Published:** 2022-11-26

**Authors:** Stephanie Jansen, Renke Lühken, Michelle Helms, Björn Pluskota, Wolf Peter Pfitzner, Sandra Oerther, Norbert Becker, Jonas Schmidt-Chanasit, Anna Heitmann

**Affiliations:** 1Department of Arbovirology and Entomology, Bernhard Nocht Institute for Tropical Medicine, 20359 Hamburg, Germany; 2Faculty of Mathematics, Informatics and Natural Sciences, University of Hamburg, 20148 Hamburg, Germany; 3Research Group Vector Control, Bernhard Nocht Institute for Tropical Medicine, 20359 Hamburg, Germany; 4Kommunale Aktionsgemeinschaft zur Bekämpfung der Schnakenplage e. V. (KABS), 67346 Speyer, Germany; 5Institute of Dipterology (IfD), 67346 Speyer, Germany; 6Institute of Global Health, Medical Faculty Heidelberg, University of Heidelberg, 69117 Heidelberg, Germany

**Keywords:** Sindbis virus, vector competence, *Aedes* and *Culex* mosquitoes, saliva titration

## Abstract

Transmission of arthropod-borne viruses (arboviruses) are an emerging global health threat in the last few decades. One important arbovirus family is the *Togaviridae*, including the species *Sindbis virus* within the genus *Alphavirus*. Sindbis virus (SINV) is transmitted by mosquitoes, but available data about the role of different mosquito species as potent vectors for SINV are scarce. Therefore, we investigated seven mosquito species, collected from the field in Germany (*Ae. koreicus*, *Ae. geniculatus*, *Ae. sticticus*, *Cx. torrentium*, *Cx. pipiens* biotype *pipiens*) as well as lab strains (*Ae. albopictus*, *Cx. pipiens* biotype *molestus*, *Cx. quinquefasciatus*), for their vector competence for SINV. Analysis was performed via salivation assay and saliva was titrated to calculate the amount of infectious virus particles per saliva sample. All Culex and Aedes species were able to transmit SINV. Transmission could be detected at all four investigated temperature profiles (of 18 ± 5 °C, 21 ± 5 °C, 24 ± 5 °C or 27 ± 5 °C), and no temperature dependency could be observed. The concentration of infectious virus particles per saliva sample was in the same range for all species, which may suggest that all investigated mosquito species are able to transmit SINV in Germany.

## 1. Introduction

Arthropod-borne viruses (arboviruses) are an emerging global health threat, demonstrated by epidemics of Yellow fever virus, dengue virus, Zika virus and chikungunya virus (CHIKV) in the last decades. One of the most important arbovirus families is the family *Togaviridae* including the genus *Alphavirus* that is characterized by a single-stranded, positive-sense RNA genome ranging from 9.7 to 11.8 kb. The genome of alphaviruses encodes 10 different proteins. The non-structural proteins (nsP1 to nsP4) and the structural proteins (capsid, E3, E2, 6K/TF, and E1). Virions are spherical, enveloped particles about 70 nm in diameter [1]. Sindbis virus (SINV) and CHIKV are important alphaviruses, known to have caused large epidemics since the 1950s [2]. First isolation of SINV took place in 1952 in Egypt, from a pool of field-caught mosquitoes [3].

SINV circulating is maintained in an enzootic cycle between mosquitoes and birds, which are the primary and amplifying hosts. Humans and other mammals are dead-end hosts [4]. As with other alphaviruses, human infection with SINV can cause arthritic disease with rash, headache, muscle pain, fatigue and occasionally fever [5]. The diseases is named as Pogosta disease in Finland, Karelian fever in Russia and Ockelbo disease in Sweden [6]. Most patients recover within weeks, but studies after an outbreak in Finland in 2013 revealed that 39% patients suffer from debilitating arthralgia and myalgia for more than 6 months [7,8]. The first human infection in Europe was noticed in the 1960s in Sweden [9] and recently, in 2021, there was a large Pogosta disease epidemic in Finland, with several hundred human cases [10]. Recent studies in South Africa revealed SINV-specific IgMs in hospitalized patients with arthralgia, meningitis and headache [11]. Beside human infections, there are occasional reports of neurologic disease caused by a SINV infection in horses and wildlife in South Africa [12,13]. Recent studies also detected SINV-neutralizing antibodies in Swedish horses, indicating infection of horses also in the Northern hemisphere [14]. SINV is widely distributed in the Old World and Australia. There are six different genotypes of SINV described: Lineage I has been isolated in Europe, Africa and the Middle East; Lineage II and VI have been isolated in Australia; Lineage III and IV have been isolated in Asia and the Middle East and Lineage V has been isolated in New Zealand [15,16]. Only SINV lineages I is associated with disease in humans [16]. Phylogenetic analysis of the different SINV I strains in Europe proposes one introduction of SINV I in northern Europe and three introductions of SINV I in southern Europe, one from Central Africa and two from northern Europe, resulting in a recombination of these strains in central/southern Europe [16]. Interestingly, there are only reports of human illness in South Africa and Fennoscandia, even if Lineage I is also widespread in the Middle East, the entirety of Europe and almost the whole of Africa. Studies in Israel demonstrated SINV-specific antibodies in humans, despite the fact that no outbreaks or clinical cases were reported so far [17]. In Germany, SINV was first isolated in 2009 from mosquito pools collected in southwestern Germany [18]. The circulation of SINV in Germany was confirmed by a study on German birds. In that study SINV was isolated from a hooded crow in Berlin. This strain was phylogenetically closely related to the strain that isolated from mosquitoes in 2009 [19]. More recently, isolation of SINV from mosquitoes and one bird (common wood pigeon) from other regions in Germany was successful [20,21,22]. Analysis of the SINV strain isolated from the common wood pigeon revealed a genetically distinct strain from the strain isolated from the hooded crow; hence, at least two different strains are circulating in Germany [22].

*Culex* mosquitoes are considered the primary vector of SINV, such as *Culex torrentium* and *Culex pipiens* biotype *pipiens*; however, a variety of genera/species collected in the field have been tested positive for SINV, such as *Culex univittatus*, *Culex neavei*, *Culiseta morsitans*, *Mansonia africana* and *Aedes* spp. [3,23,24,25]. The following mosquito species were tested positive for SINV in Germany: *Cx. torrentium; Culex pipiens sensu lato* and *Anopheles maculipennis sensu lato* [18,20]. Studies on the vector competence of SINV are scarce. However, they revealed transmission of SINV by *Cx. torrentium*, *Cx. pipiens, Aedes albopictus* and *Aedes aegypti* and no transmission of SINV by *Aedes vexans* [26,27,28]. All studies were only carried out under one temperature condition (26 °C or room temperature), which leaves the question open, whether transmission of SINV is influenced by temperature. The impact of temperature on vector competence is caused by mainly two different effects [29]. First, replication of viruses increases with higher temperatures, therefore transmission also increases with higher temperatures. Second, the antiviral immune response (RNAi) of the mosquito is decreasing through lower temperatures, leading again to higher transmission rates [30]. The balance of these two temperature-dependent effects is different for each virus and mosquito species combination.

To understand the dynamics of SINV infection and transmission in mosquitoes we investigated seven different mosquito species including two biotypes of *Culex pipiens*, lab strains as well as field-caught mosquitoes from Germany. Mosquitoes were incubated at up to four different temperature profiles and analyzed via salivation assay at up to three timepoints. Infection rate, transmission rate, transmission efficiency and the SINV-titer in the saliva was also determined.

## 2. Materials and Methods

### 2.1. Collection and Rearing of Mosquitoes

Lab strains of *Culex pipiens* biotype *molestus* (established in 2011 from egg rafts collected in Heidelberg, Germany), *Ae. albopictus* (established in 2016/17 from eggs in Freiburg, Germany) and *Culex quinquefasciatus* (a long-established colony from Bayer, Leverkusen, Germany) were reared in an insectary at 26 °C with a relative humidity of 70% and a 12:12 light:dark photoperiod, including twilight of 30 min. Eggs of *Aedes geniculatus* (49°31′ N, 8°40′ E) and *Aedes koreicus* (50°03′ N, 8°16′ E/50°05′ N, 8°16′ E/50°08′ N, 8°17′ E) were collected with ovitraps in Western Germany in 2019/2021 and reared under the same conditions as the lab strains in the insectary. Taxonomic identification was performed with larvae of the 4th stage [31,32]. Egg rafts of *Cx. torrentium* and *Cx. pip.* biotype *pipiens* were collected in the field in northern Germany (Lon: 53.467821/Lat: 9.831346) in 2018. Larvae were reared from single egg rafts at room temperature; F0 adults were reared in the insectary. Species differentiation of the *Culex* larvae was performed via molecular identification. DNA was extracted from 3 to 5 larvae per egg raft; analysis was performed via multiplex quantitative real-time PCR (qRT-PCR) as previously described [33]. *Aedes sticticus* was collected in Western Germany in 2019 (49°49′ N, 8°24′ E/49°49′ N, 8°25′ E/49°12′ N, 8°24′ E) as adults with Encephalitis Vector survey traps (EVS traps; BioQuip products, Rancho Dominguez, CA, USA), 1.5 kg dry ice was added as attractant, and transported to the insectary. For field-collected mosquitoes, 10 randomly selected mosquitoes per species were tested by pan-Flavivirus-, pan-Alphavirus- and pan-Orthobunyavirus-PCR to exclude natural infections [34,35,36].

### 2.2. Experimental Infection of Mosquitoes

Initially, 4–14 days-old female mosquitoes were starved for 24 h to reach high feeding rates. Mosquitoes were fed at room temperature by artificial blood meal containing SINV (lineage I, BNI-10865, GenBank MF 589985.1, passage six) which was isolated from a mosquito in Germany. Blood meal was performed either via blood-soaked cotton sticks for 2–3 h, with feeding rates (FR, number of engorged females per number of fed females) of 58% for *Cx. pip.* biotype *molestus*, 87% for *Cx. quinquefasciatus*, 58% for *Cx. pip.* biotype *pipiens* and 49% for *Cx. torrentium*, or via blood drops (two 50 µL drops at the bottom of the vials) for one hour, with a feeding rate of 71% for *Ae. albopictus*, 65% for *Ae. koreicus*, 74% for *Ae. sticticus* and 64% for *Ae. geniculatus*. Virus stock was propagated using Vero cells (*Chlorocebus sabaeus*; CVCL_0059, obtained from ATCC, Cat# CCL-81) and concentrated with PEG-it virus precipitation solution (System Biosciences, Palo Alto, CA, USA). Blood meal contained 50% human blood (expired blood bags), 30% fructose (8% solution), 10% filtrated bovine serum (FBS) and 10% virus solution, final virus concentration was 10^7^ plaque forming units per milliliter (PFU/mL). Subsequently, mosquitoes were sorted and the fully engorged females were placed in a new vial. Incubation of mosquitoes was executed at fluctuating temperature profiles of 18 ± 5 °C, 21 ± 5 °C, 24 ± 5 °C or 27 ± 5 °C (mimicking the fluctuating temperatures between day and night) with a relative humidity of 70% for 5 and/or 14 days, only for the lab strain *Cx molestus* additionally for 28 days. A fructose-soaked cotton pad with 8% fructose solution was continuously provided and the fructose solution was refreshed every 2–3 days. For most combinations of species/temperature/incubation period at least two independent experiments were executed and results are combined. Only for *Ae. koreicus* and *Ae. sticticus* were not enough specimens available to perform more than one experiment per timepoint/temperature.

### 2.3. Analysis of Infection and Saliva Titration

After incubation, mosquitoes were analyzed for infection and transmission. Salivation assay was performed as previously described [37]. In short, mosquitoes were anesthetized with CO_2_, immobilized and the proboscis was placed into a filter tip containing 10 µL phosphate-buffered saline (PBS). After an incubation of 30 min, filter tips with saliva were pipetted into a tube containing 10 µL PBS. Saliva-PBS-solution was pipetted into the media of Vero cells seeded in a 96-well plate to measure the cytopathic effect (CPE). To be able to estimate the amount of infectious virus particles per saliva-sample, we titrated each sample using a dilution series from 1:10 to 1 × 10^7^. Five days after infection cells were screened for CPE and supernatant was tested via qRT-PCR as described by Jöst et al. [18] including the VetMAX™ Xeno™ Internal Positive Control (Applied Biosystems, Thermo Fisher Scientific Corporation, Waltham, MA, US) according to the manufacturer’s protocol. Through this, transmission rate (TR, the number of SINV-positive saliva per number of SINV-positive bodies) and transmission efficiency (TE, the number of SINV-positive saliva per number of fed females) were calculated. The average virus titer of saliva was calculated in the following way: If only the first well showed CPE, the amount of infectious virus particles was calculated as 1 or more, but less than 10 (>1). If two wells showed CPE the number of infectious virus particles was calculated as 10 or more, but less than 100 (>10). Subsequently, definition was 3 wells > 100, 4 wells > 1000, 5 wells > 10,000, 6 wells > 100,000 and 7 wells > 1,000,000. Bodies were homogenized, RNA was isolated and qRT-PCR was performed with the abovementioned protocol. Through this, the infection rate (IR, the number of SINV-positive bodies per all fed mosquitoes) and the mean body virus titer was calculated. All negative values of the 95% values for mean body and mean saliva were set up to zero, as negative values make no biological sense. The correlation between the mean body titer and the mean saliva titer for the tested mosquito populations, temperatures and days post infection were analyzed with Spearman’s rank correlation (function cor.test in the program R (R Foundation for Statistical Computing: Vienna, Austria)) [38].

### 2.4. Survival of Infected/Uninfected Culex pipiens biotype molestus at Four Temperatures

Infection and incubation was performed as described before. Uninfected mosquitoes were also blood-fed on day zero containing 50% blood, 30% fructose solution, 10% FBS and 10% DMEM. Only fully engorged females were incubated. Survival of uninfected and SINV-infected mosquitoes was documented weekly until 28 dpi. Survival rates were calculated by Kaplan–Meier, using PRISM, version 9.3.1 (Graph Pad Software, San Diego, CA, USA). Comparison of survival rates was completed by log-rank (Mantel–Cox) test. To address the problem of multiple comparisons, the Bonferroni correction method was applied to each pair of groups.

## 3. Results

### 3.1. Vector Competence Studies

All investigated mosquito species were able to transmit SINV at all tested temperatures (Table 1). With an infection rate (IR) of 100% for all timepoints/temperature profiles *Cx. pip.* biotype *molestus* was highly susceptible for SINV infection. The transmission efficiency (TE) for SINV was higher than 50% for all investigated timepoints/temperatures. At the higher temperature profiles of 27 +/− 5 °C (henceforth referred to as 27 °C) and 24 +/− 5 °C (henceforth referred to as 24 °C) the TE was higher at the early timepoints with 97%/96% 5 dpi, compared to 60%/51% 14 dpi. At 21 °C +/− 5 °C (henceforth referred to as 21 °C) the TE was stable above 90% at both timepoints. Meanwhile, at the lowest temperature profile of 18 +/− 5 °C (henceforth referred to as 18 °C), an increase between 5 dpi (77%) and 14 dpi (100%) could be observed. The IR of *Cx. quinquefasciatus* varied between 40% and 82%, with the highest IRs at 24 °C (82% 5 dpi; 65% 14 dpi). The TE values increased from the lowest temperature profile of 18 °C (TE: 3%/10%) to around 20% at 21 °C and 27 °C for both timepoints, and the highest values of 42%/28% at 24 °C. *Culex torrentium* showed a high susceptibility for SINV infection with IRs higher than 90% for all timepoints/temperature profiles. At the highest temperature profile of 27 °C, the TE was higher at the early timepoint of 5 dpi (83%) than at the later timepoint (65%, 14 dpi). A relatively constant TE in the seventies was observed for 24 °C (73%/79%). While at the lower temperature profile of 21 °C and 18 °C the TE was clearly higher 14 dpi than 5 dpi (18 °C: 20% 5 dpi and 84% 14 dpi; 21 °C: 40% 5 dpi and 93% 14 dpi). With an IR between 27% and 77%, *Cx. pip.* biotype *pipiens* showed the lowest TE values, ranging between 0% and 37%. At 18 °C, the TE was increasing by time (7% 5 dpi, 13% 14 dpi), however, for all other temperature profiles the TE was decreasing by time (21 °C: 37% to 0%; 24 °C: 33% to 17%; 27 °C: 17% to 7%). *Aedes albopictus* showed an IR of 73% up to 100%. The TE was higher at the later timepoint of 14 dpi, with values of 67% to 80% (compared to 27–57% 5 dpi) at all investigated temperature profiles.

Because there were not enough specimens of *Ae. koreicus* available, only two temperature profiles were investigated (21 °C and 27 °C) at one timepoint (14 dpi). Likewise, the number of investigated specimens per timepoint is also lower than for all the species described before (*n* = 2, 21 °C; *n* = 15, 27 °C) and the validity of the rates for 21 °C is low, since only two individuals were investigated. *Aedes koreicus* was susceptible for SINV infection at both temperatures, with values at 27 °C of an IR of 60% and a TE of 27% (27 °C). Experiments with *Ae. sticticus* and *Ae. geniculatus* were only performed at one temperature profile and one timepoint (14 dpi), again with a low number of specimens due to shortage of available mosquitoes. Both species showed SINV-positive saliva, *Ae. sticticus* with a TE of 19% at 24 °C and *Ae. geniculatus* with a TE of 14% at 21 °C.

### 3.2. Determination of the Number of Infectious Virus Particles per Saliva Sample

Titration of saliva was successful for all species, mean values were ranging between 0.5 up to 2.98 log10 infectious virus particles per saliva sample (Table 1). *Culex pip.* biotype *molestus* showed, in general, high saliva titers, including the highest mean value from all samples at 21 °C, 5 dpi, with a mean of 2.98 (2.38–3.59) log10 infectious virus particles per mosquito. High mean titers above 2.00 were observed at the lower temperature profiles of 18 °C and 21 °C; at 27 °C mean saliva titers were below 2.00. Mean saliva titers were increasing over time at 18 °C from 2.20 at 5 dpi to 2.93 at 14 dpi, at the three higher temperature profiles the mean titers decreased by time (21 °C: 2.98 5 dpi to 2.33 14 dpi; 24 °C: 2.15 to 1.00; 27 °C: 1.76 to 1.28). Lower mean saliva titers were observed for *Cx. quinquefasciatus*, ranging from 0.50 to 1.59. Mean saliva titers slightly increased by time at the three lower temperature profiles (18 °C: 0.50 to 0.83; 21 °C: 1.00 to 1.21; 24 °C: 1.13 to 1.59) and were stable at 27 °C (0.67 for both timepoints). *Culex torrentium* showed high mean saliva titers ranging between 1.63 to 2.55 with an inconsistent pattern in relation to temperature and time. The only mean high saliva titer above 2.00 for *Cx. pip.* biotype *pipiens* was observed 5 dpi at 18 °C with 2.25. All other mean titers were in the moderate range between 1.00–2.00. With 1.13 to 2.33 *Ae. albopictus* showed mean saliva titers in the moderate up to higher values. Saliva titers were increasing by time (18 °C: 1.13 to 1.55; 21 °C: 1.95 to 2.02; 24 °C: 1.15 to 2.33), only for the highest temperature a slightly decrease was observed (27 °C: 1.57 to 1.40). For *Ae. koreicus* and *Ae. Geniculatus*, moderate mean saliva titers were observed (*Ae. koreicus*: 1.50 and 1.00; *Ae. geniculatus*: 1.50). *Aedes stictus* showed a high mean saliva titer with 2.50.

Statistical analyses revealed a statistically significant positive correlation (Spearman’s *r* 0.52, *p* = 0.0001) for the mean body titer and the mean saliva titer for the tested mosquito populations, temperatures and days post infection (Figure 1; Appendix A).

### 3.3. Investigation of Clearance of SINV Infection on the Example of Culex pipiens biotype molestus

#### 3.3.1. Survival of *Culex pipiens* biotype *molestus* for 28 Days

To study the vector competence for SINV four weeks after infection, we performed experiments with the lab strain *Cx. pip.* biotype *molestus*. Due to our observation of varying survival rates (SR) at different temperatures, we documented the survival of SINV-infected and non-infected *Cx. pip.* biotype *molestus* mosquitoes over the whole time period (Appendix A). In general, the mortality increased with increasing temperatures in both groups, culminating in a survival rate (SR) of 0.0% at 27 °C (Table 2). There was no difference in survival comparing infected and uninfected mosquitoes at the three higher temperature profiles, only at 18 °C the survival of the uninfected females was significantly higher (*p* > 0.0001) (Appendix A).

#### 3.3.2. Vector Competence of *Culex pip*. biotype *molestus* 28 dpi

All specimens incubated at 27 °C died during the experiment duration of 28 days, regardless of whether mosquitoes were infected or uninfected, hence no salivation assay was performed. For all other temperature profiles *Cx. pip.* biotype *molestus* showed SINV-positive saliva (Table 2). Transmission efficiency was highest at the lowest temperature of 18 °C with 81%, followed by 71% at 21 °C and 43% at 24 °C.

## 4. Discussion

Transmission of SINV was observed for all seven investigated mosquito species. Our findings are in agreement with previous studies, where *Cx. torrentium* mosquitoes from Sweden were demonstrated to transmit SINV in a salivation assay and *Ae. albopictus*, *Ae. aegypti* and *Cx. pip.* biotype *pipiens* being able to infect chicken with SINV [26,27]. Overall, *Cx. pip.* biotype *molestus* reached the highest transmission efficiency for all timepoints/temperature profiles with values over 50%, whereas *Cx. pip.* biotype *pipiens* and *Cx. quinquefasciatus* showed the lowest transmission efficiency for all timepoints/temperature profiles under 50%. *Aedes albopictus* reached transmission efficiencies from 27% up to 80%. With relatively low numbers of investigated specimens of *Ae. sticticus*, *Ae. koreicus* and *Ae. geniculatus* the validity of the rates is limited, but our results clearly demonstrate that all three species are able to transmit SINV. Thus, mosquitoes from the genus *Culex* as well as from the genus *Aedes* are able to transmit SINV under laboratory conditions. Interestingly, Modlmaier et al. could not detect any dissemination (legs) or transmission (saliva) with SINV after oral infection of *Ae. vexans* from Germany [28]. Due to successful infection of *Ae. vexans* with intrathoracic inoculation, they proposed that the midgut barrier impedes infection.

Transmission was observed at all temperatures and timepoints (except *Cx. pip.* biotype *pipiens* at 21 °C, 14 dpi). Transmission of SINV is temperature-independent in a range of 18 +/− 5 °C–27 +/− 5 °C. Temperature-dependent transmission patterns are often described for flaviviruses such as West Nile virus, Zika virus or Japanese encephalitis virus [39,40,41,42], showing higher transmission rates at tropical temperatures in comparison to moderate temperatures. Viruses within the *Alphavirus* genus, such as CHIKV, do not show the same clear pattern. Vector competence studies with *Ae. koreicus* infected with CHIKV clearly showed a temperature dependence [43,44], but studies on *Ae. albopictus* infected with CHIKV clearly did not show a temperature-dependent transmission from 18 to 27 °C [45]. Therefore, temperature-dependent transmission depends on the combination of species and arbovirus, and seems to be even more differentiated on mosquito populations and virus strains [46]. In our case, we detected only temperature-independent transmission of SINV for all the investigated mosquito species.

Transmission cycles of arboviruses are highly complex. Apart from the successful infection of a mosquito after a bloodmeal, including the overcoming of the midgut barrier and the salivary gland barrier, the number of virus particles in the saliva, associated with the number of inoculated infectious virus particles, plays an important role in infection of a host. There are two parameters that determine the virus concentration in the saliva, either the amount of viral RNA isolated from saliva solution or FTA cards calculated via PCR [47,48] or the amount of infectious viral particles. Since only the infectious virus particles are important for infection, measuring the viral RNA can overestimate the infectivity of the saliva. To determine the number of infectious virus particles in mosquito saliva, a plaque assay can be performed [49] or a simple titration on cells [50,51], such as our approach in this study. The amount of detected infectious Sindbis particles in our study is in the same range as presented by Dubrelle et al. and Mercier et al. for CHIKV, as well as Smith et al. for Venezuelan equine encephalitis virus, another member of the family *Togaviridae* [49,50,51]. It must be clarified that all techniques of saliva isolation are not equivalent reflecting the inoculation of saliva by a mosquito to a host. Besides the different time span of salivation, the inoculation site of the body-part may also play an important role, which is hard to mimic in a salivation assay. Turell et al., for example, showed that there is a difference in infection whether mosquitoes inoculate the saliva extravascular or directly into the vascular system [52]. In the study of Smith et al., they concluded that the amount of transmitted VEEV is less in vivo in comparison to a capillary tube, even if the time span was the same [50]. Nevertheless, the estimation of transmitted virus particles by in vitro methods allows us to assess the risk of successful arbovirus transmission. The detected saliva titers in this study are in the same range as presented by Dubrelle et al., as well as Smith et al., and it can be assumed that the number of viral particles in the saliva of all investigated mosquito species is high enough for host infection after inoculation into the host [49,50]. The positive correlation of body and saliva titer is quite reasonable; thus, it can be presumed that the body SINV titer of field collected mosquitoes gives an indication of their ability to transmit SINV. However, this is not true for all virus species. USUV-infected *Cx. pip.* mosquitoes of both biotypes showed an USUV body titer in the same range, but transmission of USUV was significantly higher for *Cx. pip.* biotype *pipiens* than for *Cx. pip.* biotype *molestus* [53].

The findings of our study, that transmission of SINV can occur at all investigated species’ temperature combinations, suggests that this is one reason for its widespread distribution in the Old World and Australia. As all investigated mosquitoes have high viral titers in the saliva and we assume they are able to transmit SINV, the question remains why there are no reported human cases in Germany. Human infection is only reported from SINV lineage 1, but there are different clades described [16]. Meanwhile, SINV strains from South Africa belong to clade C and E, SINV strains from Scandinavia belong to clade A. The SINV strains described for Germany and which we used in this study, also belong to clade A. Therefore, the absence of detected human cases in Germany cannot be explained by the circulation of different clades of SINV lineage 1. The pathogenicity of Alphaviruses is often related to the surface glycoprotein E, which is also true for SINV [54]. A single mutation in the *E1* gene of CHIKV increased the infectivity in *Ae. albopictus* mosquitoes [55]. Ling et al. investigated the differences in the *E2* gene of SINV, which encodes for a glycoprotein that is linked to the pathogenicity [54], and found no evidence of any differences in these genes for the different clades [16]. With our results, we hypothesize that the reason for the limited pattern of human disease is also not due to the occurrence of a specific vector. Further investigation into the different genotypes of SINV lineage 1 as well as ecological studies on the host population should be completed.

Our observation of increased mortality of *Culex* mosquitoes with increasing temperature is in line with other studies [56]. The impact of arbovirus infection on fitness and longevity of mosquitoes is again specific for the combination of arbovirus and mosquito species. Whilst some studies clearly showed a negative impact by arbovirus infection on mosquito survival [42], others observed varying effects. Studies with the same strain of Eastern equine encephalomyelitis virus revealed a higher mortality caused by virus infection for *Coquillettidia pertubans* but no influence on survival for *Ae. albopictus* [57]. The same was shown for CHIKV infection, which has a negative effect on *Ae. albopictus* survival, but no influence on the longevity of *Ae. aegypti* [58]. Infection with the Batai virus decreases the survival of *Ae. detritus*, but has no influence on the longevity of *Ae. aegypti* or *Cx. pipiens* [59]. However, we did not observe a negative impact by long-term SINV infection on the survival of *Cx. pip.* biotype *molestus* mosquitoes at 27 °C, 24 °C and 21 °C. This is in accordance with the study of Khoo et al., who did not detect a negative effect on the longevity of *Ae. aegypti* by SINV infection at 28 °C (28 dpi) [60]. A slight, but significant, negative effect on survival was observed at 18 °C, but it remains questionable as to whether this would have any biological effect.

Looking at the TE values over the time, there are different patterns. For *Ae. albopictus*, TEs are increasing at all temperature profiles from 5 dpi to 14 dpi. However, for *Cx. pip.* biotype *molestus* as well as *Cx. torrentium* there is an increasing TE at 18 °C, a relatively constant value at 21 °C as well as 24 °C for both timepoints, and a decreasing TE at 27 °C from 5 dpi to 14 dpi. This raises the question, whether there is a persistent SINV infection or a clearance of infection. To investigate this, we performed infection studies with *Cx. pip.* biotype *molestus* for 28 days. No clearance of SINV infection could be observed at any temperature, all TEs were slightly lower 28 dpi compared to 14 dpi (24 °C: 51% 14 dpi to 43% 28 dpi; 21 °C: 94% to 71%; 18 °C: 100% to 81%), but mosquitoes were clearly able to transmit SINV. Due to the mortality of *Cx. pip.* biotype *molestus* at a high temperature of 27 °C, no infection could be observed for this temperature. Based on these results, we hypothesize that SINV is able to establish persistent infections in mosquitoes. Persistent infections are especially important for areas with cold seasons, such as middle and northern Europe. Either the host, in the case of SINV-infected birds, or mosquitos need to develop a persistent infection to maintain the transmission cycle. Our hypothesis is in line with studies of Bergmann et al. where they found an IR of 4.6% for SINV in hibernating *Culex* mosquitoes from the field during winter 2018–2019 [61].

## 5. Conclusions

SINV can be transmitted by *Culex* as well as *Aedes* mosquitoes with high TEs. All the tested mosquito species from Germany are potent vectors for SINV and we suggest a persistent infection and persistent ability to transmit SINV, at least for *Culex* mosquitoes. Why there are no human cases observed in areas such as Germany or Israel, even if SINV is regularly detected, remains unclear. Precisely because of this ambiguity, regular surveillance programs for human SINV infections should be performed and cases of arthralgia with unknown causes should be tested for SINV.

## Figures and Tables

**Figure 1 viruses-14-02644-f001:**
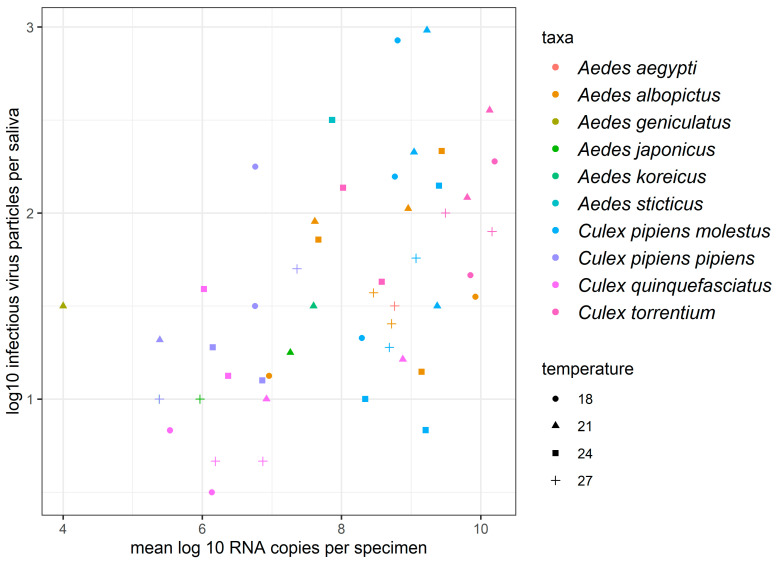
Correlation of mean body titers and mean saliva titers per specimen.

**Table 1 viruses-14-02644-t001:** Results of vector competence studies with Sindbis virus. Infection rates (IR), mean body titer, transmission rates (TR), transmission efficiency (TE) and mean saliva titer 5- or 14-days post infection (dpi). *n* = number of investigated specimens. * = only 34 of 53 saliva samples were titered. ** = only 16 of 21 saliva samples were titered. n.a. = not applicable.

Species	Temperature	dpi	*n*	IR (%)	Mean Body Titer log10 Copies/Mosquito (95% Confidence Interval)	TR (%)	TE (%)	Mean Saliva Titer log10 Infectious Virus Particle/Mosquito (95% Confidence Interval)
*Culex pipiens* biotype *molestus*	18 +/− 5 °C	5	30	100	8.77 (8.70–8.83)	77	77	2.20 (1.46–2.93)
		14	35	100	8.80 (8.59–9.02)	100	100	2.93 (2.50–3.36)
	21 +/− 5 °C	5	30	100	9.23 (8.94–9.51)	97	97	2.98 (2.38–3.59)
		14	31	100	9.04 (8.74–9.35)	94	94	2.33 (1.91–2.75)
	24 +/− 5 °C	5	55	100	9.40 (9.12–9.69)	96	96	2.15 (1.61–2.68) *
		14	35	100	8.34 (8.12–8.56)	51	51	1.00 (0.61–1.39)
	27 +/− 5 °C	5	36	100	9.07 (8.94–9.20)	97	97	1.76 (1.41–2.10)
		14	30	100	8.69 (8.40–8.98)	60	60	1.28 (0.70–1.86)
*Culex quinque-fasciatus*	18 +/− 5 °C	5	30	57	6.14 (5.24–7.03)	6	3	0.5 (n.a.)
		14	30	40	5.54 (4.29–6.78)	25	10	0.83 (0.00–2.27)
	21 +/− 5 °C	5	30	40	6.92 (5.18–8.66)	50	20	1.00 (0–43–1.57)
		14	30	43	8.88 (7.50–10.26)	54	23	1.21 (0.33–2.09)
	24 +/− 5 °C	5	50	82	6.37 (5.80–6.95)	51	42	1.13 (0.65–1.60) **
		14	40	65	6.02 (5.30–6.75)	42	28	1.59 (0.96–2.23)
	27 +/− 5 °C	5	30	60	6.87 (5.69–8.05)	33	20	0.67 (0.24–1.10)
		14	30	53	6.19 (4.94–7.44)	38	20	0.67 (0.24–1.10)
*Culex torrentium*	18 +/− 5 °C	5	30	100	9.85 (9.24–10.46)	20	20	1.67 (0.12–3.21)
		14	31	97	10.20 (9.46–10.94)	87	84	2.28 (1.43–3.13)
	21 +/− 5 °C	5	36	97	9.81 (9.02–10.60)	41	40	2.08 (0.95–3.22)
		14	40	100	10.13 (9.90–10.36)	93	93	2.55 (2.03–3.07)
	24 +/− 5 °C	5	30	97	8.02 (7.47–8.58)	76	73	2.14 (1.56–2.71)
		14	30	93	8.58 (7.95–9.21)	85	79	1.63 (1.14–2.12)
	27 +/− 5 °C	5	30	97	10.17 (9.43–10.89)	86	83	1.90 (1.29–2.51)
		14	31	94	9.49 (9.14–9.85)	69	65	2.00 (1.30–2.70)
*Culex pipiens* biotype *pipiens*	18 +/− 5 °C	5	30	33	6.76 (4.95–8.57)	20	7	1.50 (0.00–14.21)
		14	30	63	6.76 (5.23–8.29)	21	13	2.25 (0.00–5.53)
	21 +/− 5 °C	5	30	70	5.39 (4.65–6.13)	52	37	1.32 (0.33–2.31)
		14	12	42	7.09 (4.87–9.31)	0	0	n.a.
	24 +/− 5 °C	5	30	77	6.86 (5.84–7.89)	43	33	1.10 (0.33–1.87)
		14	52	37	6.15 (5.39–6.91)	47	17	1.28 (0.77–1.80)
	27 +/− 5 °C	5	30	27	7.36 (4.71–10.01)	63	17	1.70 (0.00–3.74)
		14	36	50	5.38 (3.97–6.80)	13	7	1.00 (0.00–7.35)
*Aedes albopictus*	18 +/− 5 °C	5	30	90	6.96 (5.94–7.98)	30	27	1.13 (0.13–2.12)
		14	30	73	9.92 (9.53–10.31)	91	67	1.55 (0.92–2.18)
	21 +/− 5 °C	5	38	97	7.62 (6.93–8.30)	30	29	1.95 (1.26–2.65)
		14	29	86	8.96 (7.99–9.92)	84	72	2.02 (1.32–2.72)
	24 +/− 5 °C	5	30	97	9.15 (8.57–9.74)	59	57	1.15 (0.84–1.46)
		14	30	90	9.44 (8.74–10.14)	89	80	2.33 (1.71–2.95)
	27 +/− 5 °C	5	30	100	8.46 (7.78–9.14)	47	47	1.57 (0.65–2.49)
		14	31	94	8.72 (8.22–9.22)	72	68	1.40 (1.00–1.81)
*Aedes koreicus*	21 +/− 5 °C	14	2	100	7.60 (0.00–52.79)	100	100	1.50 (−11.21–14.21)
	27 +/− 5 °C	14	15	60	5.38 (3.90–6.87)	44	27	1.00 (0.00–2.59)
*Aedes sticticus*	24 +/− 5 °C	14	16	69	7.90 (7.01–8.72)	27	19	2.50 (0.00–7.47)
*Aedes geniculatus*	21 +/− 5 °C	14	9	33	4.00 (3.53–4.47)	33	11	1.50 (n.a.)

**Table 2 viruses-14-02644-t002:** Vector competence and survival of *Cx. pip.* biotype *molestus* infected with SINV or uninfected 28 days post feed. Survival rates (SR) and in brackets the number of surviving mosquitoes per the number of fed mosquitoes, Infection rates (IR), mean body titer (log10 FFU/mosquito), transmission rates (TR) and transmission efficiency (TE). n.a. = not applicable.

Species	Temperature	Infection Status	SR 28 Days Post Feed (%)	IR (%)	Mean Body Titer log10 FFU/Mosquito (95% Confidence Interval)	TR (%)	TE (%)
*Culex pipiens* biotype *molestus*	18 +/− 5 °C	SINV	47.9 (57/119)	100	8.29 (8.18–8.41)	81	81
		None	73.0 (92/126)	n.a.	n.a.	n.a.	n.a.
	21 +/− 5 °C	SINV	9.6 (17/178)	100	9.38 (9.21–9.54)	71	71
		none	9.2 (12/130)	n.a.	n.a.	n.a.	n.a.
	24 +/− 5 °C	SINV	5.5 (7/127)	100	9.21 (8.93–9.49)	43	43
		none	0.0 (0/158)	n.a.	n.a.	n.a.	n.a.
	27 +/− 5 °C	SINV	0.0 (0/124)	n.a.	n.a.	n.a.	n.a.
		none	0.0 (0/155)	n.a.	n.a.	n.a.	n.a.

## Data Availability

The data presented in this study are available on request from the corresponding author.

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
