# Peer review of "Vector Competence of Mosquitoes from Germany for Sindbis Virus"

_viruses, 2022, doi:10.3390/v14122644_

Round 1

Reviewer 1 Report

Well described paper. I have no critical comment. 

Author Response

Thank you for reviewing the paper

Reviewer 2 Report

The manuscript by Dr. Jansen and colleagues describes extensive studies on the vector competence of eight mosquito species occurring in Germany for a Sindbis virus (SINV) lineage 1 isolate also from Germany. They did this using four different temperature profiles (for most of the mosquito species) in order to see whether or not infection rate, transmission rate, transmission efficacy or the virus titre in the mosquitoes’ saliva are somehow correlated with the temperature. The outcome is interesting although not very surprising, but has not been done with such a comprehensiveness. All mosquito species investigated were capable to transmit SINV in a temperature-independent manner.

This is really interesting and adds to our knowledge of this alphavirus. Unfortunately, it does not explain why no SINV-infections are observed in Germany yet, but that may come when temperatures are continue to rise.

Minor issues for consideration:
lines 45 and 54 is a mixture of singular and plural used, pls. correct.
Pls. provide a reference for the sentence in lines 100-101.
106: transmission (twice not in capital).
129: Orthobunyavirus-PCR (h and virus missing).
132 onwards: Pls. use consistent writing when describing % (either always with a blank after the number, or never).
143: Pls. use bags instead of preservation.
166-167: Pls. delete “, PCR protocol was performed” otherwise this sentence reads odd.
169: … per number…
171: …per number…
190: Pls. mention how you calculated the survival rate in the PRISM software (Kaplan-Meier?)
Fig. 1: I am not sure I completely understand this figure. According to the legend, it correlates the mean log 10 RNA copies of saliva with that of the body of the mosquito. I don’t understand why Spearman rank correlation was used instead of Pearson as the scale would allow this. This would be more accurate. Further it is not visible which dot relates to what species or temperature as everything in uniformed in the scatter plot. Would it make sense to use different colours or icons in order to make the difference with regard to temperature or species visible? Why didn’t the authors add a trend line?
359: There is no such thing like an E protein in alphaviruses! There are E1-E2 heterodimers on the surface of the virion, pls. correct!
There is one point in the discussion missing although it might be a little bit speculative. Do the authors think that the promiscuity of SINV with regard to mosquito vector species is a (or the) reason it is geographically the most widespread alphavirus.

Author Response

lines 45 and 54 is a mixture of singular and plural used, pls. correct.
> Changed accordingly

Pls. provide a reference for the sentence in lines 100-101.
> Reference is added (Adelman et al 2013)

106: transmission (twice not in capital).
> Changed accordingly

129: Orthobunyavirus-PCR (h and virus missing).
> Changed accordingly

132 onwards: Pls. use consistent writing when describing % (either always with a blank after the number, or never).
> Changed accordingly

143: Pls. use bags instead of preservation.
> Changed accordingly

166-167: Pls. delete “, PCR protocol was performed” otherwise this sentence reads odd.
> Changed accordingly

169: … per number…
> Changed accordingly

171: …per number…
> Changed accordingly

190: Pls. mention how you calculated the survival rate in the PRISM software (Kaplan-Meier?)
> Yes, survival rates were calculated by Kaplan-Meier, information is added.

Fig. 1: I am not sure I completely understand this figure. According to the legend, it correlates the mean log 10 RNA copies of saliva with that of the body of the mosquito. I don’t understand why Spearman rank correlation was used instead of Pearson as the scale would allow this. This would be more accurate. Further it is not visible which dot relates to what species or temperature as everything in uniformed in the scatter plot. Would it make sense to use different colours or icons in order to make the difference with regard to temperature or species visible? Why didn’t the authors add a trend line?

> We used Spearman coefficient as this also works with monotonic relationships and does not necessarily require a  linear relationship as the Pearson coefficient. There might be also a threshold effect, i.e. non-linear behaviour in the relationship between the body and saliva titre. For the same reason we also did not include a trend line in the figure. However, we added different colours and shapes to indicate the different species and temperatures.

359: There is no such thing like an E protein in alphaviruses! There are E1-E2 heterodimers on the surface of the virion, pls. correct!
> I guess this comment is meant for line 408-4011, which is changed from the protein level to the gene level.

There is one point in the discussion missing although it might be a little bit speculative. Do the authors think that the promiscuity of SINV with regard to mosquito vector species is a (or the) reason it is geographically the most widespread alphavirus.

> Interesting point. Yes, we think that this might be a reason for the distribution on five continents. We added this point to the discussion in line 397-399.

Reviewer 3 Report

The article is well written, the methods are clearly presented, and the results are interesting and support the conclusion. I have only one mention. Culex pipiens pipiens and Culex pipiens molestus are biotypes of the same species as you mentioned. So, it cannot be said that you investigated the vector competence for eight species, but only for seven, with the mention that in the case of the Culex pipiens species, both biotypes were tested.

Congratulation for this interesting paper!

Author Response

Thank you for reviewing the paper. 

As you mentioned, we changed from eight species to seven species, including two biotypes.